# Intraspecific Morphometric Variation in a New Species of *Ceratomyxa* Thélohan 1892 (Cnidaria) from the South Atlantic Ocean: An Ecomorphological Study Using Geometric Morphometrics [note 1]

**DOI:** 10.3390/biology14010079

**Published:** 2025-01-16

**Authors:** Delfina M. P. Cantatore, Martina Lisnerová, Paula S. Marcotegui, María A. Rossin, Astrid S. Holzer

**Affiliations:** 1Laboratorio de Ictioparasitología, Instituto de Investigaciones Marinas y Costeras (IIMyC), Facultad de Ciencias Exactas y Naturales (FCEyN), Universidad Nacional de Mar del Plata (UNMdP)-Consejo Nacional de Investigaciones Científicas y Técnicas (CONICET), Funes 3350, Mar del Plata 7600, Buenos Aires, Argentina; pmarcotegui@hotmail.com (P.S.M.); alejandrarossin@gmail.com (M.A.R.); 2Institute of Parasitology, Biology Centre of the Czech Academy of Sciences, Branišovská 31, 37005 České Budějovice, Czech Republic; lisnerova@paru.cas.cz; 3Fish Health Division, University of Veterinary Medicine, Veterinärplatz 1, 1210 Vienna, Austria

**Keywords:** *Ceratomyxa fialai* n. sp., myxospore shape variation, Myxozoa, Argentina

## Abstract

We describe a new species of microscopic parasitic cnidarian infecting the Argentine croaker, a fish native to the South Atlantic Ocean, contributing to the understanding of parasite biodiversity in the Argentine Sea. For the first time, we efficiently apply landmark-based geometric morphometrics (GM) to analyze the shape and size of these spores, uncovering natural variation driven by developmental noise and phenotypic plasticity. Our findings further suggest that variations in spore shape may influence their dispersal capacity in marine environments. This research offers novel insights into the ecological and evolutionary processes shaping disease dynamics in marine environments.

## 1. Introduction

Myxozoa Grassé 1970 is a highly diverse and widely distributed group of microscopic parasitic cnidarians characterized by a simplified body plan consisting of only a few cells [1,2], initially classified as protozoans due to their simplified morphology and parasitic lifestyle. These spore-forming parasites have complex life cycles, alternating between a myxospore stage in intermediate hosts (vertebrates, mainly fish) and an actinospore stage in definitive hosts (annelids or bryozoans) [3]. To date, more than 2600 species have been described [4], primarily from freshwater and marine environments [1,2]. However, their actual diversity is likely underestimated, suggesting many more species remain to be discovered [4].

*Ceratomyxa* Thélohan 1892 is one of the largest myxozoan genera, comprising about 285 nominal species [5,6,7,8,9,10,11,12,13,14,15,16,17,18,19,20]. Species of *Ceratomyxa* are characterized by elongated, crescent-shaped, or arcuate, but occasionally subspherical or oval, mature myxospores with conical or sub-hemispherical valves, which significantly exceed half the axial diameter of the myxospore [2,7]. Beyond these qualitative descriptions of shape, the morphological characterization of *Ceratomyxa* myxospores relies on sets of isolated linear measurements (i.e., length, width, and thickness) as well as the posterior angle of the spore. Typically, these metrics are reported with averages alongside measures of variability, such as standard deviation and range [5,10,21,22]. Therefore, morphological descriptions and comparisons within and between species have been based on qualitative and quantitative descriptors and performed using univariate and multivariate statistics within the framework of traditional morphometrics (TM), which can limit the full representation of shape complexity. Geometric morphometrics (GM), in contrast, applies advanced multivariate statistical techniques, offering a more powerful alternative framework for representing, describing, and studying morphologies in a more comprehensive and integrated manner than the TM approach [23,24,25]. Although GM has proven its versatility in numerous biological studies [26], it has rarely, if ever, been applied to myxozoan.

Intraspecific morphometric variation is a common phenomenon in myxozoans [27]. Similarly to other myxozoans, *Ceratomya* species show significant intraspecific variation in myxospore morphology [6,22,28,29]. This variation has been partly attributed to deformation due to myxospore lability [30]. However, understanding the factors that drive variability in the size and shape of myxospores is essential for interpreting the role of these important taxonomic traits in myxozoan ecology and evolution, partly because of their adaptive importance [31,32,33]. The realized phenotype, revealed in the final shape of an organism, is a complex phenomenon [34]. Indeed, it is widely accepted that phenotypic expression results from the balance between sources of variation, such as genetic factors, environmental effects, and developmental errors, and the counteracting regulatory mechanisms buffering against them [34,35], resulting in distinct forms differing from each other in diverse ways. Understanding these interactions is critical for comprehending the full scope of morphological diversity within *Ceratomyxa* and other myxozoan species.

The aim of the present study is to evaluate the utility of the landmark-based geometric morphometrics (GM) approach and apply it as a tool to describe, quantify, and test patterns of intraspecific variations in the size and shape of myxospores from a new species of *Ceratomyxa* within an ecomorphological framework. Additionally, this study uses an integrative taxonomic approach, combining morphological, bioecological, and molecular data to describe a new species. This characterization is supported by the analysis of novel SSU rDNA data, along with data from other *Ceratomyxa* species, to explore the phylogenetic relationships of the new species within the *Ceratomyxa* clade.

## 2. Materials and Methods

### 2.1. Fish and Parasite Collection and Processing

Fourteen specimens of the Argentine croaker Berg 1895, ranging from 25.0 to 32.0 cm in total length and 200 to 518 g in weight, were captured by commercial trawling along the coast of Buenos Aires Province (37°15′ S, 56°58′ W), Argentine Sea, in September 2016. These fish were obtained from commercial fishing, and, as they were already deceased at the time of collection, ethical approval was not required. During necropsy, the fish were examined for the presence of lesions and/or cysts in tissues and organs under a stereoscopic microscope. The gall and urinary bladders were removed, and their contents were withdrawn using sterile syringes. Wet mounts of an aliquot of bile and urine and squash preparations of livers and kidneys were examined by light microscopy for the detection of vegetative stages and/or mature myxospores. Tissue samples of parasitized fish containing plasmodia and/or myxospores were then subdivided and fixed in 96% ethanol for DNA sequencing or directly used for morphological characterization.

### 2.2. Digital Microphotographs Acquisition

Digital microphotographs of myxospores on agarose pads as a substrate were taken at 100× magnification under differential interference contrast (DIC) using a Leica DM2500 stereoscopic microscope equipped with a Leica DFC295 camera (Wetzlar, Germany). The image stacks were saved at a resolution of 300 dpi and in .tif format, retaining scale and size information.

A large number of myxospores were photographed, and a couple of photographs of the same spore were taken to ensure that sufficient spores in adequate orientation and high-resolution microphotographs could be selected. A total of 107 microphotographs of different myxospores obtained from five fish host specimens (20, 16, 19, 17, and 35 myxospores from each fish host specimens, respectively) were selected for traditional and landmark-based descriptions, geometric morphometric analyses, and downstream multivariate statistical hypothesis testing.

### 2.3. Traditional Morphometric Description

Myxospores were measured in ImageJ v.1.45 s [36], following the general guidelines of [21]. In addition, the posterior spore angle of the ceratomyxids was measured as described by [22]. All measurements are given in micrometers as the mean ± standard deviation (SD), followed by the range in parentheses, unless otherwise stated. The morphological description was performed using spores obtained from the same five samples that were used for DNA extraction and SSU rDNA sequencing.

### 2.4. Landmark-Based Geometric Morphometrics (GM)

For GM analyses, only myxospore shell valves were chosen, since internal structures such as polar capsules, sporoplasm, and nuclei are movable elements. Such structures often introduce systematic errors and bias the entire GM analysis because their position relative to the rest of the landmark configuration can vary considerably. Therefore, our digitized data contain information on a two-dimensional configuration structured from eight defined landmarks that provide a decent coverage of the outline of the myxospores and efficiently capture ceratomyxid shape variation. To assess whether the newly defined landmark configuration can be used to produce reproducible results in GM analyses, the average repeatability (RPT) [37] of each shape coordinates of all replicates was estimated.

The software tpsUtil v1.78 [38] was used to organize the images and convert them into a TPS file format. TPS images were digitized in tpsDig2 v2.31 [38] to acquire landmarks’ coordinates by the first author, who was also responsible for taking the photographs in order to reduce bias in the digitization process. To perform some of the subsequent analyses (i.e., to calculate RPT and % ME and to obtain the symmetric component of the shape), the shape of the 107 photographed *Ceratomyxa* spores was digitized twice (landmark labeling), rendering two replicates of the dataset; the two sets were labeled at one-week intervals, over a three-day period. The remaining procedures and analyses were performed with R (R Core team 2020) in RStudio (RStudio Team 2020), using the packages factoextra v1.7.0 [39], geomorph v3.3.1 [40], Morpho v2.8 [41], rptR 0.9.22 [37], and RRPP v0.6.1 [42,43].

A generalized Procrustes analysis (GPA) [44,45] was performed on landmark configurations to remove the parameters of non-shape variation (i.e., differences in size, location, and orientation). Therefore, the resulting aligned landmark configurations (Procrustes coordinates) contain only shape information. The software tpsSmall v1.29 [38] was used to assess if the Euclidean distance in the tangent space could be considered an approximation of the Procrustes distances among the Procrustes coordinates obtained, allowing them to be used in subsequent multivariate statistical analyses.

The proportion of the total phenotypic variance observed in the dataset that can be attributed to non-biological sources relative to “Imaging” (i.e., differences in image acquisition of the same myxospore due to mounting) and “Digitizing” (i.e., differences in the placing of the landmarks on the same photograph) should be quantified before conducting GM analyses. Note that it is unlikely to mount and photograph the same myxospore more than once; therefore, only errors occurring from the digitization of landmarks could be estimated. To quantify this bias, the measurement error (% ME) [46,47] for shapes was estimated based on the Procrustes ANOVA by decomposing the total phenotypic variation into within-group (biological) and among-group (bias in landmark placement) components. Moreover, to identify and exclude aberrant configurations (defined as those specimens that fall above the upper quartile) in the dataset as a consequence of a high measurement error or mislabeling of landmarks, specimens were ordered according to the Procrustes distance from the mean shape.

Procrustes shape coordinates were subsequently used to assess *Ceratomyxa* n. sp. morphospace by principal component (PC) analysis (PCA). The number of PCs regarded as meaningful was obtained by comparing the ratio of a PC and its successor with a threshold based on a log-likelihood ratio [48]. The proportional contribution of each landmark to the variation explained by each PC was calculated for PC1 and PC2 to estimate which landmark contributed more than the average to the variation. Convex hulls in PCA were used to visualize the portion of the morphospace occupied by myxospores parasitizing different fish host specimens. Additionally, deformation grids were displayed on the X and Y axes, illustrating the extreme myxospore shapes of each PC axis as an estimate divergence in the relative location of landmarks from their configuration in the average specimen.

Myxospores of the genus *Ceratomyxa* exhibit bilateral object symmetry, as defined by [49], in which the plane of bilateral symmetry passes through the middle of the entire spore (suture line) and separates it into connected mirror images of each other (shell valves). In specimens that exhibit symmetric patterning, the shape tangent space for the entire landmark configuration can be decomposed into separate and orthogonal, complementary subspaces [50]. These subspaces account for shape variation among myxospore specimens (subspace of completely symmetric shape variation) and shape variation between the shell valves of each myxospore (subspace of totally asymmetric shape changes). Studies on the variation of symmetric structures need to circumvent inherited statistical problems, considering the symmetry and unambiguously separate variation among specimens from intra-individual asymmetries [50]. Therefore, to assess whether myxospore mean shapes differ among infected hosts when controlling for myxospore size, or whether myxospore shapes are host-individual-dependent in their response to size, the symmetric component of shape was extracted from the Procrustes coordinates [40]. Myxospore size was calculated from landmark configurations as the centroid size, the square root of the summed squared distances from the configuration centroid [51]. The extracted symmetric component of the shape was then used as the dependent variable in a one-way Procrustes ANCOVA [52], with a residual randomization permutation procedure (RRPP) using the sequential (type I) sums of squares based on Procrustes distance (10,000 iterations for significance testing) [53]. Additionally, a one-way Procrutes ANOVA approach was performed to statistically assess whether myxospore size (as the centroid size) differed among host specimens, with the same settings as the analysis above. The use of permutation tests to obtain *p* values in Procrustes ANOVA/ANCOVA designs does not circumvent the problem that these kinds of statistical tests are sensitive to the heterogeneity of variances among groups. For this reason, permutational tests of multivariate dispersion were used to check the homogeneity of the dispersion of the samples from their group centroids [54].

### 2.5. Molecular and Phylogenetic Analyses

DNA was extracted from the five infected bile subsamples preserved in ethanol at 4 °C. Each ethanol-fixed sample was centrifuged twice at 5000 rpm for 3 min and washed with 1 mL of DNAse-free water and the ethanol was removed. DNA was extracted from the final pellet using a QIAgen DNeasy^®^ Blood & Tissue Kit (animal tissue protocol) (Qiagen Inc., Hilden, Germany) following the manufacturer’s protocol, and the purified DNA was used as template DNA for subsequent polymerase chain reactions (PCR).

Partial SSU rDNA sequences of *Ceratomyxa* species were obtained by assembling overlapping parts amplified with both universal and *Ceratomyxa*-specific primers. Thereby, the reactions with the universal eukaryotic ERIB1 (5′-ACCTGGTTGATCCTGCCAG-3’) and ERIB10 (5′-CTTCCGCAGGTTCACCTACGG-3’) primers (both [55]) in the first PCR were followed by nested PCR with the 18S-cerF (5′-CTWGTTGGTADGGTAGTG-3’) and 18S-cerR (5′-GTACAAGAGGCAGAGACGTAT-3’) primers (both [56]). PCRs of the SSU rDNA were conducted in 20 µL of reaction volumes, comprising: 1 µL template DNA (1.3–16.9 ng µL^−1^), 20 pmol of each primer, 200 µM of dNTPs, 1× EasyTaq^®^ buffer with a final MgCl_2_ concentration of 1.5 mM, and 1 unit of EasyTaq^®^ DNA polymerase in ultrapure water. Reactions were performed in a programmable thermal cycler (Techne ^3^Prime thermal cycler) with the PCR cycling parameters used for the primary/nested PCR set as follows: initial denaturation at 95 °C for 5 min, followed by 30 cycles of amplification at 95 °C for 1 min, 48 °C/50 °C for 1 min, and 72 °C for 2 or 1 min, followed by a terminal extension at 72 °C for 10 min. All amplified PCR products were purified using the QIAquick PCR Purification Kit (Qiagen Inc., Hilden, Germany) following the manufacturer’s protocol and were sent to Macrogen Inc. (Seoul, South Korea) for bidirectional DNA sequencing using the same primers as those used for the nested PCR.

Forward and reverse sequences of SSU rDNA were aligned and assembled into consensus sequences in SeqMan II v5.05 (DNASTAR Inc., Madison, WI, USA) and were compared to other sequences available in the GenBank (GB) databases using standard nucleotide BLAST searches [57].

The SSU rDNA alignment comprised a total of 53 sequences, including the novel *Ceratomyxa* sequence, *Myxodavisia bulani* (GB acc. n° KM273030), *Palliatus indecorus* (GB acc. n° DQ377712), one sequence of an undescribed *Ceratomyxa* species (GB acc. n° DQ377699), and one of the available GB sequences of each of the 47 representatives of the genus formally described retrieved from GB. *Chloromyxum clavatum* (GB acc. n° JQ793641) and *Chloromyxum riorajum* (GB acc. n° FJ624481) were used as outgroup taxa. The alignment was created with MAFFT v7.409 [58] using E-INS-i multiple alignment methods, with a default gap opening penalty (-op = 1.53) and a gap extension penalty (-ep = 0.0) implemented in Geneious Prime 2019.0.4 [59] and manually edited by removal of ambiguously aligned regions. The resulting alignment was 2222 bp long.

Phylogenetic analyses were conducted using both maximum likelihood (ML) and Bayesian inference (BI). The maximum likelihood (ML) analysis was performed in RAxML v7.2.8 [60] using the GTR+ Γ model of nucleotide substitution which was selected as the best-fitting model of evolution in jModelTest [61]. Bootstrap support values were calculated based on 1000 replicates. The Bayesian inference (BI) analysis was conducted in MrBayes v3.0 [62], using the GTR + Γ model of evolution (six rates of substitution; gamma rate variation across sites; eight categories used to approximate the gamma distribution). MrBayes ran for one million generations, using two independent runs of four simultaneous Markov Chain Monte Carlo (MCMC) algorithms, with every 100th tree saved to estimate posterior probabilities.

## 3. Results

Mature myxospores of a coelozoic myxozoans consistent with the morphological diagnosis of the genus *Ceratomyxa* were observed floating freely in the bile of nine out of the 14 fish sampled (64%). No abnormal discoloring or thickening of the gall bladder was noted in the infected individuals. Morphological, bioecological, and molecular data supported the erection of a new species. No other myxozoan infections were found in the fish sampled.

### 3.1. Description

#### 3.1.1. Morphological Description Taxonomic Summary

Phylum: Cnidaria Hatschek, 1888

Subphylum: Myxozoa Grassé, 1970

Class: Myxosporea Bütschli, 1881

Order: Bivalvulida Shulman, 1959

Family: Ceratomyxidae Doflein, 1899

Genus: *Ceratomyxa* Thélohan, 1892

*Ceratomyxa fialai* n. sp. Cantatore and Lisnerová (Figure 1)

ZooBank number for species:

urn:lsid:zoobank.org:act:8B76B453-4905-4C2D-8251-EA925679F224

Vegetative stages: no vegetative stages were observed.

Mature myxospores (n = 20) from a single host (type): typical mature spore, crescent-shaped in the frontal view, with a convex anterior and concave posterior ends, measuring 8.6 ± 0.7 (7.4–10.1) µm in length and 20.5 ± 4.1 (14.0–29.5) µm in thickness. The shell valves were symmetrical or slightly asymmetrical, tapering to round tips with no discernable ornamentation (without mucous envelope). The sutural line was straight, prominent, running between the polar capsules. Two subspherical, equal-sized polar capsules were situated medially in the anterior part of the spore, measuring 3.5 ± 0.3 (3.0–4.2) µm in length and 3.3 ± 0.2 (2.7–3.6) µm in width. The number of turns of the polar tubule was not conspicuous. One sporoplasm with two nuclei was present, occupying almost a third of each shell valve. The posterior angle was 104.3° ± 22.8° (50.7–146.4°). Abnormal spores with three polar capsules and three shell valves were rarely observed among the spores.

Measures of myxospores (n = 87) from additional hosts (n = 4): the spore measured 8.6 ± 1.2 (6.7–11.4) µm in length and 21.8 ± 4.4 (14.0–32.5) µm in thickness. Polar capsules measured 3.1 ± 0.3 (2.0–4.0) µm in length and 3.0 ± 0.3 (2.0–3.9) µm in width. The posterior angle was 100.6° ± 24.7° (40.7–151.6°).

#### 3.1.2. Taxonomic Summary

Type host: Argentine croaker *Umbrina canosai* Berg 1895 (Sciaenidae, Perciformes).

Type locality: off the coast of Villa Gessel (37°15′ S, 56°58′ W), Buenos Aires Province, Argentine Sea.

Site of infection: lumen of the gall bladder (coelozoic).

Prevalence of infection: 64% prevalence (nine fish infected out of 14 examined).

Type material: syntype MLP-Oi4474 (Giemsa-stained slide) and voucher MLP-Oi4475 (myxospores in 96° ethanol) deposited in the Invertebrate Collection, Museo de La Plata, La Plata, Argentina.

DNA sequences: The SSU rDNA gene sequence is available in the GenBank database under the accession number PQ678412 (assembled seq).

Etymology: the species is named *Ceratomyxa fialai* n. sp. in honor of Dr. Ivan Fiala, in recognition of his lifelong commitment to the study of myxozoan parasites. His work has significantly advanced our understanding of these fascinating organisms, and his passion and dedication have inspired a generation of researchers.

#### 3.1.3. Taxonomic Affinities

Comparisons between described *Ceratomyxa* representatives and the new species were conducted considering: (i) host identity, (ii) geographical distribution, and (iii) phylogenetic relatedness (see below). In this regard, (i) *Ceratomyxa fialai* n. sp. can be differentiated from the four *Ceratomyxa* species parasitizing the gall bladder of sciaenids, namely *C. aggregata* Davis, 1917, *C. auratae* Rocha, Casal, Rangel, Castro, Severino, Azevedo, and Santos, 2015, *C. daysciaenae* Sarkar and Promanik, 1994, and *C. venusa* Jameson, 1931, by having larger myxospores (Table 1). Moreover, the new species has thicker myxospores than *C. aggregata*, *C. daysciaenae*, and *C. venusa* and bigger polar capsules than *C. daysciaenae* (Table 1). (ii) Among the *Ceratomyxa* spp. inhabiting the same geographic region (Argentine Sea) that parasitize teleost fish, *C. fialai* n. sp. can be distinguished from *C. argentina* Alama-Bermejo, Hernández-Orts, Huchon, and Atkinson, 2021, and *C. raneyae* Alama-Bermejo, Hernández-Orts, Huchon, and Atkinson, 2021, by having smaller and thicker myxospores and smaller polar capsules, from *C. dissostichi* Brickle, Kalavati, and MacKenzie, 2015, by having larger myxospores and by the shape of the polar capsules, from *C. lobata* Evdokimova, 1977, and *C. opisthocornata* (Evdokimova, 1977) by having larger and thicker myxospores, and from *C. elegans* Jameson, 1929, by having slightly bigger polar capsules (Table 1). Unfortunately, the description of *C. flexa* Evdokimova, 1977, pre-dates the molecular era and provides very few morphological features for a proper taxonomic comparison. Notwithstanding, *C. flexa* has been reported parasitizing another fish-host Order (Pleuronectiformes) and, therefore, based on the current knowledge of host specificity among *Ceratomyxa* spp. [6], host identity likely allows species differentiation. Moreover, (iii) among the 11 phylogenetically most related species of ceratomyxas revealed by the molecular analyses, namely *C. auratae* (previously compared), *C. batam* Qiao, Shao, Pengsakul, Chen, Zheng, Wu, and Hardjo, 2019, *C. cornutti* Surendram, Vijayagopal, and Sanil, 2024, *C. cyanosomae* Heiniger and Adlard, 2013, *C. gleesoni* Gunter and Adlard, 2009, *C. gurnardi* Sobecka, Szostakowska, Zietara, and Wiecaszek, 2013, *C. hallettae* Gunter, Whipps, and Adlard, 2009, *C. ostorhinchi* Heiniger and Adlard, 2013, *C. robertsthomsoni* Gunter, Whipps, and Adlard, 2009, *C. rueppellii* Heiniger and Adlard, 2013, and *C. thalassomae* Heiniger, Gunter, and Adlard, 2008, the new species can be differentiated by having larger myxospores. Moreover, *C. fialai* n. sp. has thicker myxospores than *C. cornutti* and *C. gurnardi* and wider myxospores than *C. robertsthomsoni* and *C. thalassomae*; bigger polar capsules than *C. batam*, *C. cyanosomae*, *C. gleesoni*, *C. robertsthomsoni*, *C. rueppelli*, and *C. thalassomae*; and a smaller posterior angle than all named species but *C. robertsthomsoni* and *C. thalassomae* with extremely variable posterior angles (Table 1).

### 3.2. Landmark-Based Geometric Morphometrics

High intraspecific morphological variation was observed for *Ceratomyxa fialai* n. sp., with the morphotype of the average shape shown in Figure 2.

The newly defined landmark configuration (Figure 2) showed a high degree of repeatability (R > 0.99). In addition, a very good fit between tangent space and shape space was identified using tpsSmall (*p* > 0.9), so that tangent space approximations to the underlying curved space can be used. Moreover, the Procrustes ANOVA for assessing the % ME showed that the individual variation greatly exceeded the bias from digitization; less than 1% of the total phenotypic variance was due to the imprecision of placing landmarks in the same specimen over the two replicates. These results (RPT, distances approximation, and % ME) indicate that the data are of sufficient quality to allow subsequent analyses. No potential outliers were observed in the dataset, and therefore, all 107 landmark configurations were used.

As previously described, PCs should be considered meaningful if their value is 1.32 times bigger than the value of the subsequent PC. In this sense, the first five PCs were qualified to be considered meaningful. Most of the total variance was explained by the first two PCs (82.8%), whereas the remaining meaningful PCs explained 8.6%, 4.7%, and 1.5% of total variance, respectively. PC1 accounted for 64.1% of the total variance of change in the myxospore shape based mainly on the posterior spore angle (Figure 3). PC2 represented 18.7% of the total shape variance, mainly attributable to changes in the length of the myxospores (Figure 3). Landmarks that contributed more than expected to shape differences along PC1 were landmarks 7 (32.7%), 3 (29.5%), and 1 (16.2%), whereas the ones that contributed more than average to the variation explained by PC2 were landmarks 5 (30.8%), 4 (15.6%), and 6 (15.3%). The extensive overlaps of the specimens in the PCA plot indicate that they share a common shape (occupied almost the same morphospace), as expected. Nonetheless, a subtle displacement among groups (myxospores in the gall bladder of individual host specimens) was observed on the PC2 axis that underlied the distribution of individuals in the morphospace.

The symmetric component of the shape (variation among myxospores) represented 78.1% of the total variance, while the remaining 21.9% was accounted for by the asymmetric component of shape variation with respect to the plane of bilateral symmetry (between shell valves). A one-way Procrustes ANCOVA analysis performed with RRPP revealed that the main effects, myxospores size, and host specimen were statistically significant (*p* < 0.001 for both main effects), whether the interaction between myxospore size and host identity was not (*p* > 0.05) (Table 2) (i.e., myxospore size had a significant impact on myxospore shape, and the host specimen had a significant impact on myxospore shape controlling for myxospore size); meaning that allometric patterns were the same across the host specimens. The model explained 38% of the total myxospore shape variation, 23% attributable to myxospore size and 11% attributable to host identity. Because we used sequential (type I) sums of squares, we looked at the results after reordering the terms in the model (intruding host identity first) and obtained the same result. Notwithstanding, the proportion of variation explained by each term changed, albeit slightly (21% attributable to myxospore size and 13% attributable to host identity). All measured individuals were mature spores; centroid sizes in the sample ranged from 16.1 to 32.5, with a coefficient of variation of just 15.2%. Myxospore size (as centroid size) differed among individual hosts (*p* < 0.001); the latter covariable explained 25% of myxospores size variation (Table 3). There was no evidence of violations of homoscedasticity (all *p* values > 0.05) in the analyses.

### 3.3. Molecular and Phylogenetic Analyses

We obtained partial SSU rDNA sequences for *Ceratomyxa fialai* n. sp. from each of the five isolates, whose overlapping parts were identical. The assemblage of them allowed us to generate a consensus sequence of 1575 bp of the SSU rRNA gene submitted to GenBank under the accession number PQ678412. According to the BLASTn search, the newly assembled sequence was different from all *Ceratomyxa* sequences deposited in the GenBank database (accessed 28 August 2024). Indeed, no other sequence was found with more than 92.6% similarity to the one of the new species.

Phylogenetic analyses (ML/BI) produced similar tree topologies that generally agreed with the findings of [12,56]. The analyses resolved different subclades within the *Ceratomyxa* species, with the new species clustering within subclade E (as defined by [56]) or subclade A1 (according to [12]), both of which include species parasitizing the gall bladder of marine teleosts. Within this group (subclade E), *C. fialai* n. sp. forms a well-supported sublineage alongside 11 other *Ceratomyxa* species, namely *C. auratae* (GB acc. n° KP765721), *C. batam* (GB acc. n° MF509267), *C. cornutti* (GB acc. n° ON818298), *C. cyanosomae* (GB acc. n° JX971424), *C. gleesoni* (GB acc. n° EU729693), *C. gurnardi* (GB acc. n° JQ071439), *C. hallettae* (GB acc. n° FJ204248), *C. ostorhinchi* (GB acc. n° JX971425), *C. robertsthomsoni* (GB acc. n° FJ204253), *C. rueppellii* (GB acc. n° JX971423), and *C. thalassomae* (GB acc. n° EU045332). The new species occupies a basal position in this subclade (Figure 4).

## 4. Discussion

The Argentine croaker *Umbrina canosai* Berg 1895 (Sciaenidae) is a migratory demersal fish endemic to shelf waters in the subtropical and temperate Southwestern Atlantic [72]. The parasitic fauna of this commercial fish has been recently studied for stock discrimination and tracking migration purposes [73,74]; nevertheless, no ceratomyxid species was registered parasitizing it. In this study, we report a myxozoan infection found in the bile within the gall bladder of the Argentine croaker from the Argentine sea. The myxospores found exhibit all the typical features of the genus *Ceratomyxa* [2,7] and have been successfully distinguished from its previously described congeners based on an integrative taxonomic approach that combines morphological, bioecological, and molecular analyses. We provide evidence consistent within species-level differences and, therefore, designate *C. fialai* as a new species.

Until recently, the study of myxozoans in the Argentine Sea has received little attention [20,33,75], leading to significant gaps in our knowledge of *Ceratomyxa* diversity in the Argentine Sea. The scarcity of documented species (see Table 1) and the existence of several undescribed ones (e.g., [76]) within this genus emphasizes the necessity for further investigation. Indeed, only recently, the first *Ceratomyxa* parasite in a cartilaginous fish has been recorded in the area [33]. Furthermore, only seven *Ceratomyxa* species have been described, thus far, from six teleost fish species belonging to five different orders in Argentine waters. Among these, *C. flexa* Evdokimova, 1977, was originally described from *Paralichthys patagonicus* Jordan 1889 (Paralichthyidae, Pleuronectiformes). However, the species name ‘*C. flexa*’ has been previously used for a parasite of the rubyfish *Plagiogeniom rubiginosum* (Hutton, 1875) (Emmelichthydae) from New Zealand [77], resulting in *C. flexa* Evdokimova, 1977, being a junior homonym. Therefore, we propose the replacement name *C. evdokimovae nomen novum* in accordance with articles 23 (23.1, 23.3, 23.4, and 23.6) and 60 (60.1 and 60.3) of the International Code of Zoological Nomenclature (ICZN).

The phylogenetic trees obtained in the present work were consistent with those obtained in previous molecular studies, including most of the *Ceratomyxa* species for which molecular data are available (e.g., [12,56]). We add a new sequenced species to the most recently branching taxon-rich subclade (subclade E according to [56]). Within this subclade, the new *Ceratomyxa* species shares its phylogenetic placement with other ceratomyxid parasites of nine different host families (Apogonidae, Carangidae, Labridae, Lethrinidae, Mugilidae, Serranidae, Sparidae, Triglidae, and Zanclidae) belonging to four different fish orders (Acanthuriformes, Mugiliformes, Perciformes, and Scorpaeniformes) from five geographic regions (off the coast of Arabia, Australia, Indonesia, Portugal, and Scotland), constituting a well-supported sublineage. Previous studies suggest that phylogenetic patterns are related to fish host family and to geographical locations [64,78]. However, taxonomic identity and distribution of the host do not impel the overall phylogeny of *Ceratomyxa* [64], as observed in our results. The descriptions of the congeners parasitizing sciaenids and recorded in the Argentine Sea are predominantly based on morphological studies, limiting our understanding of their evolutionary relationships. Two recently described *Ceratomyxa* species from the banded cusk eel (Ophidiidae) inhabiting the Argentine Sea [20] provide an exception; however, these species do not appear to be closely related at the molecular level, according to the phylogenetic reconstruction. This highlights the necessity for further research to comprehensively assess the diversity and distribution of these aquatic parasites. Such studies will enhance our understanding of their phylogenetic and evolutionary relationships, particularly in the underexplored South West Atlantic Ocean.

Myxozoans are dioxenous parasites with two obligatory hosts and two waterborne infective stages (myxospores and actinospores). Myxospores contain most characters used in myxozoan taxonomy, with their size and shape being key for systematics [2]. The simplicity of these microscopic spores, coupled with the limited number of measurable characters and the intraspecific morphological variability, presents challenges for the traditional morphological approach (TM), which may introduce subjectivity and overlook subtle variations in the overall shape. In this study, we used landmark-based geometric morphometrics (GM) for the first time in myxozoans to conduct a comprehensive analysis of the morphology of ceratomyxid myxospores, while examining their natural variation within and among myxospores infecting the gall bladder of five conspecific fish host specimens. By analyzing the overall geometry rather than focusing solely on disjointed measurements, we achieved greater accuracy in describing the shape of *Ceratomyxa*. This approach enabled a more detailed visualization of myxospore shape variation, capturing variability in terms of landmark coordinates and geometric deformations. The proposed two-dimensional landmark configuration can effectively be utilized to study the morphology of myxospores, specifically ceratomyxid-like morphotypes, as demonstrated by the average repeatability (RPT) and the measurement error (% ME). Nevertheless, alternative landmark configurations should be tested before studying other myxozoan morphotypes.

Accurate morphological studies are essential not only for making taxonomic decisions and assessing biodiversity, but also for understanding morphological variation within ecological and evolutionary contexts. In natural environments, intraspecific phenotypic variation arises from complex interactions among genetic, environmental, and developmental factors across different levels of biological organization (molecules, genes, cells, individuals, populations, and environments) [35]. Through landmark-based GM analyses, we effectively represented and quantified phenotypic variation at the organismal and population levels. Each observed myxospore shape of the newly described *Ceratomyxa* species diverged from the average species-specific morphology (Figure 2), representing a potential outcome of the species’ overall shape variation and offering indirect insights into its phenotypic variability [35]. Moreover, GM allows for the detection, partitioning, and quantification of symmetric and asymmetric components of shape variation [25]. This capability enables a quantitative description of (a)symmetry in organisms, and, in the case of the new species, complemented the qualitative characterization of its shell valves. Bilateral characters, such as shell valves in *Ceratomyxa* species, can exhibit true asymmetry, as observed in *C. microlepis* Azevedo, Rocha, Casal, Saõ Clemente, Matos, Al-Quraishy, and Matos, 2013 [79], or may vary due to developmental factors [80,81], resulting in random physiological differences during sporogony that lead to slight asymmetry. This asymmetry can, thus, be considered a measure of developmental noise or random molecular events [82], contributing to the observed morphological variation in myxospores, as we hypothesize is the case for the new species.

The myxospore phase (or vertebrate phase) begins with the initial infection of the permissive hosts by actinospores. This phase involves continuous cellular processes (cell proliferation), where presporogonic stages may persist and replicate in tissues and organs that differ in location from the final site of sporogony [2]. During this phase, myxozoans face challenges such as motility for invasion, migration through host tissues, and evasion of the immune system [2]. Determining whether myxospores within a single host specimen stem from the infection by one or multiple actinospores is fundamental in elucidating the extent to which clonal proliferation following a single infection (i.e., a single genotype) contributes to the observed morphological diversity within each host specimen due to morphological plasticity (phenotypic variations shaped by the in-host microenvironments). However, this determination is challenging; while the probability of a single host becoming infected by more than one actinospore appears to be low in the open sea, largely due to the observed moderate prevalence of infection, this does not completely rule out the possibility of multiple infections. Notably, the variation observed among the myxospores infecting individual fish was consistent across all host specimens (Figure 3; homoscedasticity in the analysis), suggesting that this is a species-specific trait.

In order to better understand further underlying causes of myxospore phenotypic variation under natural conditions, we applied GM methods to model myxospore shape in relation to myxospore size and host specimen. Although the proportion of total variance explained by the model was moderate (38%), our results indicate that both significantly influence myxospore shape. While the relationship between size and shape is well understood and particularly significant in aquatic microorganisms (i.e., biological functionality) [83], untangling whether host specimen serves as a proxy for genetic factors, in-host environmental factors, or a combination thereof remains challenging. Moreover, GM methods allow to model size and shape independently. The shape independence in analyzing size variations (as centroid size) can potentially reveal patterns in size variations (e.g., size CV%) across species which are essential for taxonomy, ecology, and evolutionary studies. Determining whether the observed 15% CV of size for the new species is a common trait among myxozoans, or at least within ceratomyxids, requires further application of this tool. Future studies applying the GM approach to other myxozoans will enhance our understanding of myxospore size evolution and its adaptive significance (e.g., [33]).

Regardless of the underlying causes of myxospore morphological variation, understanding the extent and patterns of variation among myxozoan spores in natural populations would offer insights into the ecological and adaptive significance of morphological traits and their variability. It is the expressed variation within a population that constitutes the raw material upon which natural selection acts and that drives potential ecological and evolutionary consequences, even in the absence of selection [84]. Natural dispersal of waterborne myxozoan stages at local geographic scales has been proposed to be influenced by several biotic and abiotic factors, which could potentially affect the encounter rate of infectious spores with receptive hosts [85]. Most likely, infective spores disperse passively, influenced by a combination of spore attributes and water current movements [86]. The morphology of the aquatic microorganisms interacts with physical forces in viscous fluids characterized by low Reynolds numbers (Re), often exhibiting adaptive traits aimed at better fitting the environmental conditions. Indeed, spore size, shape, and ornamentation are closely linked to spores dispersal capability, determining varying durations of suspension in the water column and, potentially, enhancing dispersal efficiency within the environment [87,88,89]. Myxospore shape has been associated with flotation capacity in the water column, suggesting that spores with extensive prolongations or caudal processes may exhibit higher buoyancy compared to those with smaller dimensions [90,91]. Moreover, myxospores produced in intermediate fish hosts demonstrate remarkable viability and can remain infectious to annelids for extended periods after release, ensuring prolonged transmission opportunities [86,91,92]. Therefore, both intrinsic attributes—the morphology of the spores and their persistence in the environment—significantly influence their dispersal potential and encounter rate with annelid definitives, whose distribution is often characterized as patchy.

The application of landmark-based GM in various fields of myxozoan studies, including taxonomy, biodiversity assessment, and ecological and evolutionary morphology, holds substantial promise. This tool offers an improved approach to studying shape variation in myxozoans, although our understanding of the processes driving morphological variability in myxozoans remains incomplete.

## 5. Conclusions

This study describes a new species of *Ceratomyxa* (Cnidaria) infecting the Argentine croaker, contributing to the poorly studied parasite biodiversity in the Argentine Sea. It also presents the first efficient application of landmark-based geometric morphometrics (GM) to analyze the shape and size of its myxospores, providing novel insights into their natural variation. The results reveal that spore morphology is influenced by developmental noise and phenotypic plasticity, highlighting the interplay between genetic and environmental factors. These findings underscore the potential role of spore shape in determining dispersal capabilities and ecological adaptability in marine environments.

By advancing our understanding of spore variability and its drivers, this research contributes to the broader study of parasite biodiversity and disease dynamics in the Argentine Sea. Future studies should extend the use of geometric morphometrics to explore spore shape variation in other *Ceratomyxa* species and other myxozoan morphotypes, further enriching our understanding of parasite morphology and its ecological implications.

## Figures and Tables

**Figure 1 biology-14-00079-f001:**
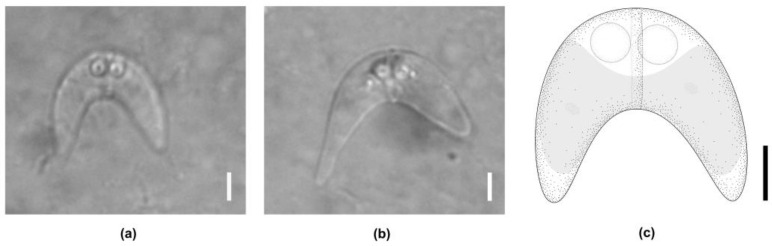
*Ceratomyxa fialai* n. sp. from the gall bladder of *Umbrina canosai* Berg 1895 (Sciaeidae). (**a**) Typical mature myxospore, sutural view, (**b**) asymmetrical mature myxospore, sutural view, (**c**) schematic drawing of a mature myxospore, sutural view. Scale bars: A, B, C = 5 µm.

**Figure 2 biology-14-00079-f002:**
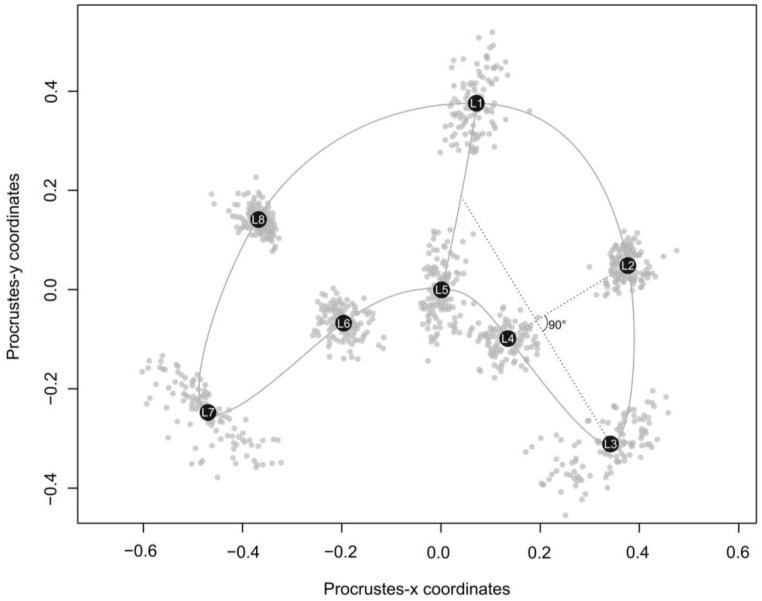
Schematic outline of *Ceratomyxa fialai* n. sp. mean shape showing patterns of variation of the Procrustes coordinates (grey circles). The set of eight geometric morphometric landmarks used in this study are shown as mean shape coordinates (black circles). Landmarks: L1: suture line’s point on the convex side of the myxospore; L2, L4, L6, and L8: geometric landmarks at the outermost (L2 and L8) or innermost (L4 and L6) margin of the myxospores generated from a perpendicular line located at the midpoint of the line connecting the middle of the suture line with L3 or L7; L3 and L7: points at the maximal curvature of the shell valve tips; and L5: suture line’s point on the concave side of the myxospore.

**Figure 3 biology-14-00079-f003:**
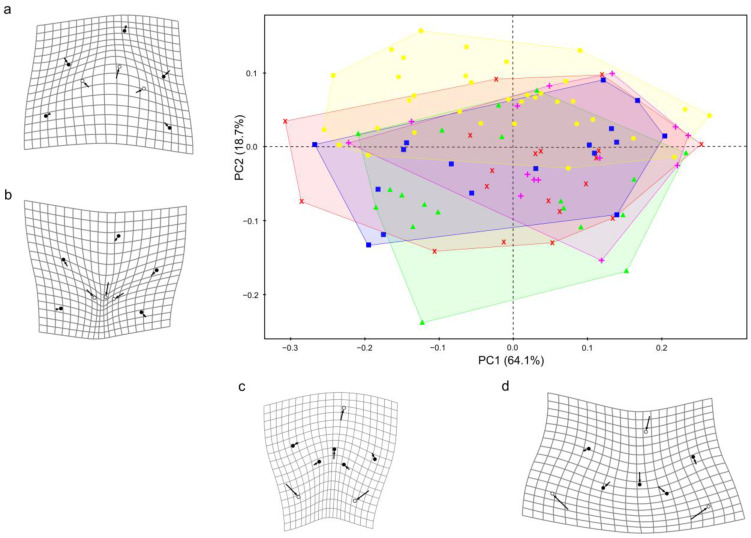
PCA plot of PC1 vs. PC2 showing the distribution of myxospores of *Ceratomyxa fialai* n. sp. (i.e., individual shapes) in a tangent space (large image). Symbols represent myxospore shapes collected from different host specimens (●, ▲, ■, +, x). Translucent-colored areas represent convex hulls, which visualize the portions of the morphospace occupied by different groups (different individual hosts). The deformation grid plots depict the shape changes from the mean shape (arrow start point) to (**a**,**c**) minimum or (**b**,**d**) maximum extreme shapes (circles) along PC1 and PC2. The open circle indicates landmarks contributing more than average to the respective PC.

**Figure 4 biology-14-00079-f004:**
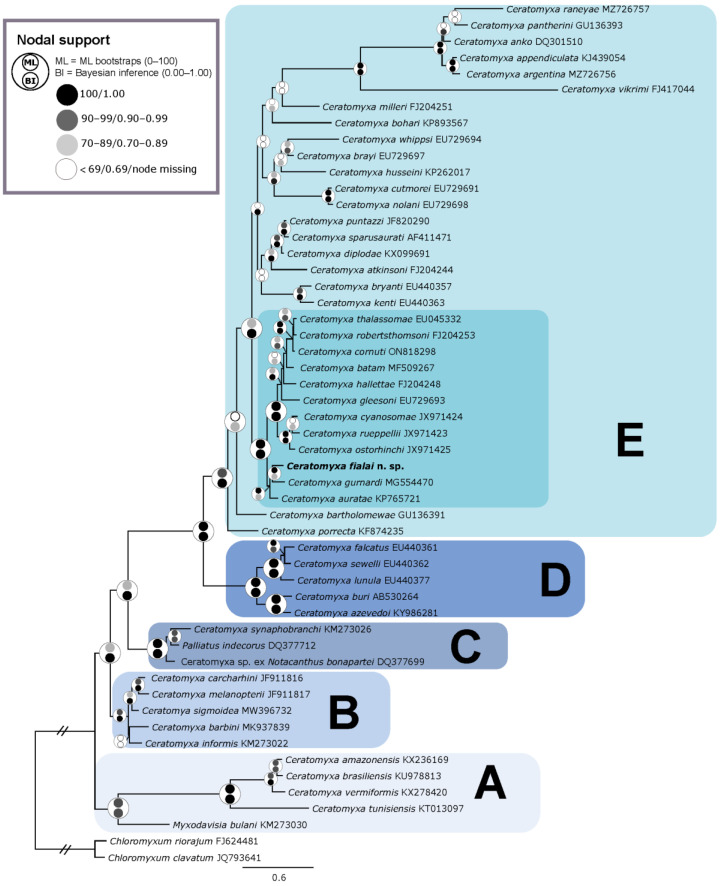
Phylogenetic tree based on SSU rDNA including *Ceratomyxa fialai* n. sp. (bold), *Myxodavisia bulani*, *Palliatus indecorus*, and closely related *Ceratomyxa* species. *Chloromyxum clavatum* and *Chloromyxum riorajum* were used as outgroup taxa. Maximum likelihood/Bayesian inference nodal supports are shown at every node in a circle colored according to the legend on the upper left side. The capital letters A–E represent the subclades proposed by [56].

**Table 1 biology-14-00079-t001:** Comparison of *Ceratomyxa fialai* n. sp. with *Ceratomyxa* species parasitizing fish from the Sciaenidae family, from the Argentine sea, and phylogenetically related species. Dimensions are given in micrometers, expressed as the mean ± standard deviation, followed by the range in parentheses, unless otherwise stated. Abbreviations: l = length, PA = posterior angle, PC = polar capsules, S = spores, V = valves, w = width, t = thickness, eq = equals, sph = spherical, pyr = pyrifom.

Species	Type Host/ Other Hosts	Locality	GenBank Acc. N°	S Size (l × t)	PC Size (l × w)	PA	Other Traits	References
*Ceratomyxa fialai* n. sp.	*Umbrina canosai* (Sciaenidae)	Off the coast of Villa Gessel, Buenos Aires, Argentina	PQ678412	8.6 ± 0.7 (7.4–10.1) × 20.5 ± 4.1 (14.0–29.5)	3.5 ± 0.3 (3.0–4.2) × 3.3 ± 0.2 (2.7–3.6)	104.3 ± 22.8 (50.7–146.4)°	V generally eq., some uneq., PC subsph.	Present study
*C. aggregata* Davis, 1917	*Leiostomus xanthurus/* *Micropogon undulatus* (Sciaenidae)	Beaufort Sea, USA	-	(6.0–7.0) × 50	3.5	180° *	PC sph.	[63]
*C. argentina* Alama-Bermejo, Hernández-Orts, Huchon, and Atkinson, 2021	*Raneya brasiliensis* (Ophidiidae)	Patagonian coast, Argentina	MZ726754 MZ726756	11.9 ± 1.1 (10.3–14.5) × 17.4 ± 2.1 (13.6–21.6)	4.5 ± 0.3 (3.7–5.1) × 4.4 ± 0.3 (3.7–5.0)	94.0–179.8°	V roughly eq., PC subsph.	[20]
*C. auratae* Rocha, Casal, Rangel, Castro, Severino, Azevedom and Santos, 2015	*Sparus aurata* (Sparidae)	Alvor estuary, Atlantic coast of Portugal	KP765721	6.7 ± 0.7 (5.3–7.6) × 27.0 ± 3.0 (19.7–31.2)	3.6 ± 0.2 (2.9–3.8) × 3.5 ± 0.3 (2.9–3.8)	167.6° *	PC subsph., five PTC	[64]
*C. batam* Qiao, Shao, Pengsakul, Chen, Zheng, Wu, and Hardjo, 2019	*Trachinotus ovatus* (Carangidae)	Batam Island, Indonesia	MF509267	3.8 ± 0.4 (2.7–4.6) × 19.2 ± 1.8 (16.2–22-0)	2.3 ± 0.2 (2.0–2.8) × 2.6 ± 0.2 (2.3–2.9)	162.3 ± 5.4 (155.4–175.2)°	V uneq., PC subsph.	[13]
*C. cornuti* Surendram, Vijayagopal, and Sanil, 2024	*Zanclus cornutus* (Zanclidae)	Lakshadweep Islands, Arabian Sea	ON818298	7.0 ± 0.4 × 26.6 ± 1.8	3.5 ± 0.2 × 3.4 ± 0.4	174.6 ± 3.6 (167.2–181.5)°	V uneq., PC uneq.	[19]
*C. cyanosomae* Heiniger and Adlard, 2013	*Ostorhinchus cyanosoma* (Apogonidaae)	Off Lizard Island, Great Barrier Reef, Australia	JX971424	6.1 ± 0.8 (5.0–8.0) × 20.0 ± 1.8 (16.7–24.2)	2.5 ± 0.2 (2.1–2.8) × 2.3 ± 0.1 (2.0–2.7)	164.2 ± 12.2 (139.2–186.8)°	V almost eq., PC sph.	[65]
*C. daysciaenae* Sarkar and Pramanik, 1994	*Daysciaena albida* (Sciaenidae)	Hooghly River estuary, West Bengal, India	-	6.0 (5.5–7.0) × 65.1 (55.0–75.0)	2.14 (1.8–3.0)		V eq., PC sph.	[66]
*C. dissostichi* Brickle, Kalavati, and MacKenzie, 2015	*Dissostichus eleginoides* (Nototheniidae)	Patagonian shelf, Argentina	-	3.8 ± 0.6 (3.2–4.5) × 17.8 ± 0.7 (15.4–22.8)	2.6 ± 0.1 (2.2–3.6) × 1.8 (1.8)	162.5° *	V eq., PC pyr., PTC 4–5	[67]
*C. elegans* Jameson, 1929	*Porichthys porosissimus*, *Triathalassothia argentina* (Batrachoididae)	Bahia Blanca and San Antonio Oeste, Argentina	-	6.9 (6.0–8.0) × 24.9 (18.0–31.0)	2.9 (2.5–3.7)		V slighlty uneq., PC sph., PTC 3–4	[68]
*C. flexa* Evdokimova, 1977 (syn *C. evdokimovae* in the present work)	*Paralichthys patagonicus* (Paralichthyidae)	North Patagonian Shelf, Argentina	-	(6.0–11.9) × (25.2–27.0)	(2.8–4.0)	Variable	PC sph.	[69]
*C. gleesoni* Gunter and Adlard, 2009	*Plectropomus leopardus* (Serranidae)	Off Heron Island and Lizard Island, Great Barrier Reef, Australia	EU729693	6.1 ± 0.5 (5.0–7.0) × 19.9 ± 1.6 (16.0–22.0)	2.3 ± 0.2 (1.5–3.0) × 2.2 ± 0.2 (1.5–2.5)	(135–180)°	PC sph., V almost eq.	[70]
*C. gurnardi* Sobecka, Szostakowska, Zietara, and Wiecaszek, 2013	*Eutrigla gurnardus* (Triglidae)	Shetland Islands, Scotland	JQ071439	5.8 ± 0.8 (4.9–7.4) × 26.4 ± 4.0 (20.3–31.1)	3.4 ± 0.4 (2.7–3.8) × 2.9 ± 0.2 (2.9–3.3)	155.6° *	PC almost sph., PTC 3	[71]
*C. hallettae* Gunter, Whipps, and Adlard, 2009	*Lethrinus harak* (Lethrinidae)	Lizard Island, Great Barrier Reef, Australia	FJ204248	4.9 ± 0.4 (3.7–5.9) × 20.5 ± 3.0 (13.9–26.5)	2.1 ± 0.4 (1.2–3.0) × 1.9 ± 0.3 (1.2–2.5)	(150–190)°	V uneq., PC sph.	[70]
*C. lobata* Evdokimova, 1977	*Odontesthes incisa* (Atherinopsidae)	North Patagonian Shelf, Argentina	-	(6.3–7.0) × (14.0–14.7)	3.5	-	PC sph.	[69]
*C. opisthocornata* (Evdokimova, 1977)	*Odontesthes incisa* (Atherinopsidae)	North Patagonian Shelf, Argentina	-	(6.4–8.0) × (9.6–14.4)	3.2	-	PC sph.	[69]
*C. ostorhinchi* Heiniger and Adlard, 2013	*Ostorhinchus aureus* (Apogonidaae)	Off Point Cloates, Ningaloo Reef, Australia	JX971425	6.8 ± 0.8 (5.3–8.6) × 24.2 ± 1.6 (21.2–27.5)	3.3 ± 0.5 (2.4–4.6) × 2.5 ± 0.4 (1.7–3.2)	162.6 ± 18.5 (135.0–204.0)°	V almost eq., PC subsph.	[65]
*C. raneyae* Alama-Bermejo, Hernández-Orts, Huchon, and Atkinson, 2021	*Raneya brasiliensis* (Ophidiidae)	Patagonian shelf, Argentina	MZ726757	11.6 ± 1.2 (9.4–14.0) × 30.6 ± 3.8 (22.3–37.4)	3.7 ± 0.4 (2.9–4.3) × 3.5 ± 0.4 (2.7–4.2)	105.9–166.7°	V roughly eq., PC subsph.	[20]
*C. robertsthomsoni* Gunter, Whipps, and Adlard, 2009	*Liza vaigiensis* (Mugilidae)	Off Lizard Island, Great Barrier Reef, Australia	FJ204253	4.7 ± 0.4 (4.0–5.9) × 17.3 ± 3.6 (12.2–24.0)	2.1 ± 0.3 (1.7–2.7) × 2.0 ± 0.3 (1.5–2.9)	(109–180)°	V uneq., PC subsph.	[70]
*C. rueppellii* Heiniger and Adlard, 2013	*Ostorhinchus rueppellii* (Apogonidaae)	Off Point Cloates, Ningaloo Reef, Australia	JX971423	6.4 ± 0.5 (5.2–7.3) × 23.6 ± 2.5 (17.3–28.3)	2.6 ± 0.4 (1.9–3.8) × 2.4 ± 0.3 (1.78–3.0)	164.2 ± 12.2 (139.2–186.8)°	V almost eq., PC subsph.	[65]
*C. thalassomae* Heiniger, Gunter, and Adlard, 2008	*Thalassoma lunare* (Labridae)	Off Heron Island and Lizard Island, Great Barrier Reef, Australia	EU045332	5.0 ± 0.7 (3.3–6.4) × 18.9 ± 1.2 (16.4–22.2)	2.9 ± 0.2 (2.2–3.3) × 2.8 ± 0.2 (2.2–3.0)	Slighlty concave to straight	V almost eq., PV subsph.	[22]
*C. venusa* Jameson, 1931	*Cynoscion nobilis* (Sciaenidae)	North Pacific Ocean, USA	-	(4–6) × (4–6)	-	Almost strait	V eq.	[5]

* obtained from drawings in the original description.

**Table 2 biology-14-00079-t002:** One-way Procrustes ANCOVA statistics based on the randomized residual permutation procedure (RRPP) with 10,000 random permutations. Model: myxospore shape~myxospore size * host identity. *df* = degrees of freedom, MS = mean squares, Z = effect size.

Source	*df*	MS	Pseudo *F*	Z	*p*
Myxospores size	1	0.56	35.36	3.53	<0.001
Host identity	4	0.07	4.40	2.65	<0.001
Myxospores size:host identity	4	0.03	1.62	1.02	0.15
Residuals	97	0.02			

**Table 3 biology-14-00079-t003:** One-way Procrustes ANCOVA statistics based on randomized residual permutation procedure (RRPP) with 10,000 random permutations. Model: myxospore size~host identity. *df* = degrees of freedom, MS = mean squares, Z = effect size.

Source	*df*	MS	Pseudo *F*	Effect Size (Z)	*p*
Host identity	4	88.26	8.60	2.99	<0.001
Residuals	102	10.26			

## Data Availability

The data presented in this study are publicly accessible in GenBank under the accession number PQ678412 and in ZooBank under record ID urn:lsid:zoobank.org:act:8B76B453-4905-4C2D-8251-EA925679F224 and Publication: urn:lsid:zoobank.org:pub:39229BA7-F2EC-4147-B6E6-9D94E4275AD5.

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
