# Peer review of "Intraspecific Morphometric Variation in a New Species of *Ceratomyxa* Thélohan 1892 (Cnidaria) from the South Atlantic Ocean: An Ecomorphological Study Using Geometric Morphometrics [Author-notes fn1-biology-14-00079]"

_biology, 2025, doi:10.3390/biology14010079_

Round 1
Reviewer 1 Report
Comments and Suggestions for Authors
The manuscript reports a new species of Ceratomyxa, microscopic parasitic cnidarian from the gall bladder of the Argentine croaker in the Argentine sea, and the new species was confirmed by identifying diagnostic morphological characteristics, molecular analyses based on SSU rDNA sequence. To my knowledge, this is the first to apply landmark-based geometric morphometrics in myxozoan research, providing a detailed analysis of conspecific morphometric variation. I thoroughly enjoyed reviewing this manuscript and only have some minor requests for revision.
1. In Abstract, the main subject of this manuscript to report a new species, Ceratomyxa fialai n. sp.. The information of Ceratomyxa flexa replaced by Ceratomyxa kimovae is not necessary to present here.
2. In section 3.1.2 Taxonomic summary. Type host was present, but there is no additional host`s information.
3. In Fig 1 a and b, the photos were used to show the morphological characteristics, but the quality of photo is low. Please replace with new photos.
4. In section 3.3, the explanation of Figure 4, showing the phylogenetic tree in this study was not clear. Please rephase the sentences.
Author Response
On behalf of all co-authors, I thank the anonymous reviewers for their constructive comments, which helped us strengthen this work.
Comment 1: In Abstract, the main subject of this manuscript to report a new species, Ceratomyxa fialai n. sp.. The information of Ceratomyxa flexa replaced by Ceratomyxa evdokimovae is not necessary to present here.
Response 1: We appreciate the reviewer’s observation regarding the content of the Abstract. The sentence mentioning Ceratomyxa flexa replaced by Ceratomyxa evdokimovae has been removed, as suggested. This adjustment ensures the focus remains solely on the main subject of the manuscript.
Comment 2: In section 3.1.2 Taxonomic summary. Type host was present, but there is no additional host`s information.
Response 2: We appreciate the reviewer’s observation regarding the host information in section 3.1.2. As per taxonomic conventions, the "Type host" field in the taxonomic summary includes only the scientific name, common name, and taxonomic position of the host. Additional details about the host, such as its size range, weight, and capture location, are provided in section 2.1 ("Fish and parasite collection and processing"). Furthermore, further ecological and distributional context of the host is discussed in section 4 ("Discussion"). We believe this structure adheres to standard practices in taxonomic descriptions but would be happy to make further adjustments if necessary.
Comment 3: In Fig. 1 a and b, the photos were used to show the morphological characteristics, but the quality of photo is low. Please replace with new photos.
Response 3: We appreciate the reviewer’s suggestion regarding the quality of the photos in Figures 1a and 1b. Unfortunately, the current equipment available in our laboratory limits the resolution of the images. While we made every effort to capture the most representative photographs of the morphological characteristics, we are unable to obtain higher-quality images with the resources at hand. We hope the current images are still adequate for illustrating the key features of the new species, and we are open to any further suggestions on how to improve the presentation.
Comment 4: In section 3.3, the explanation of Figure 4, showing the phylogenetic tree in this study was not clear. Please rephase the sentences.
Response 4: Thank you for your valuable feedback. In response to your suggestion, we have rephrased the explanation of Figure 4 to enhance clarity. We hope the revised description better conveys the phylogenetic relationships and the position of Ceratomyxa fialai n. sp. within the tree. All modifications have been tracked using the "Track Changes" feature for your convenience.
Reviewer 2 Report
Comments and Suggestions for Authors
The paper represents an important contribution by incorporating integrative taxonomy for the description of a new species of an interesting and poorly studied group of parasites. The authors carried out good research in a well manuscript. I only have 4 recommendations to improve the work:
1. Introduction. Please add a brief mention about the previous taxonomic placement of Myxozoa as protozoans (lines 44-45).
2. Results. Mention if there were any other parasites in the review of the hosts (trichodines?, monogeneans?). Additionally, put the length of each of the 5 fish where the respective quantities of myxospores were found (line 114).
3. Results. Please, explain in more detail and more explicitly the ecomorphological referents of the results obtained. The morphogeometric analysis clearly shows the morphospaces but the ecological (ecomorphological) implication is not so clear.
4. Results. Please explicitly mention the meaning of the capital letters in the phylogenetic tree obtained (fig. 4), particularly their relationship to fish taxa.
Author Response
On behalf of all co-authors, I thank the anonymous reviewers for their constructive comments, which helped us strengthen this work.
Comment 1: Introduction. Please add a brief mention about the previous taxonomic placement of Myxozoa as protozoans (lines 44-45).
Response 1: Thank you for your suggestion. In response to your comment, we have added a brief mention about the previous taxonomic placement of Myxozoa as protozoans. The information has been incorporated into the introduction, as it is crucial to understand the historical context of their classification.
Comment 2: Results. Mention if there were any other parasites in the review of the hosts (trichodines?, monogeneans?). Additionally, put the length of each of the 5 fish where the respective quantities of myxospores were found (line 114).
Response 2: Thank you very much for your insightful comment. We appreciate your suggestion to include information about other potential parasites in the hosts, such as trichodines and monogeneans. As a more comprehensive review of the parasitic fauna in these hosts has been previously presented by Canel et al. (2019, 2021), our study focused specifically on documenting the presence of Ceratomyxa fialai in the gall bladder of the fish. Regarding the size of the fish, all the specimens examined were adult, and no specific analysis of their size was conducted in this study. For this reason, we believe that including the host size data might not be essential for the scope of this manuscript. However, we agree that it could be an interesting avenue for future research to assess biological aspects such as the presence of other parasites, parasite load, host size and sex, and how these factors might influence the variability of myxospore characteristics.
Comment 3: Results. Please, explain in more detail and more explicitly the ecomorphological referents of the results obtained. The morphogeometric analysis clearly shows the morphospaces but the ecological (ecomorphological) implication is not so clear.
Response 3: Thank you for your thoughtful comment. While the results section primarily presents the morphogeometric analysis and the identification of morphospaces, we have explored the ecomorphological implications of these findings more extensively in the discussion section. Ecomorphology seeks to understand how morphological traits relate to the ecological roles and adaptations of organisms. In our study, the variability observed in myxospore morphology is connected to ecological factors, particularly the potential dispersal of spores in the environment. This aspect is important because it helps explain how morphological variation may reflect ecological pressures and influence the spread and infection capacity of myxospores across different habitats. While the results focus on the morphological patterns, the discussion provides a more detailed ecomorphological interpretation, linking these variations to the ecological significance of myxozoan spores. We believe this interpretation aligns with the goals of ecomorphology by shedding light on the adaptive value of these morphological traits in response to ecological factors. We hope this helps clarify how the ecomorphological perspective was addressed in our work.
Comment 4: Results. Please explicitly mention the meaning of the capital letters in the phylogenetic tree obtained (fig. 4), particularly their relationship to fish taxa.
Response 4: Thank you for your suggestion. We have updated the figure legend to include an explicit mention of the meaning of the capital letters (A-E) in the phylogenetic tree. These letters correspond to the subclades proposed by Fiala et al. (2015) [56], as requested. The updated legend now clearly reflects this relationship.